# Photodynamic Activity of TMPyP4/TiO_2_ Complex under Blue Light in Human Melanoma Cells: Potential for Cancer-Selective Therapy

**DOI:** 10.3390/pharmaceutics15041194

**Published:** 2023-04-09

**Authors:** Mihaela Balas, Simona Nistorescu, Madalina Andreea Badea, Anca Dinischiotu, Mihai Boni, Andra Dinache, Adriana Smarandache, Ana-Maria Udrea, Petronela Prepelita, Angela Staicu

**Affiliations:** 1Department of Biochemistry and Molecular Biology, Faculty of Biology, University of Bucharest, 91-95 Splaiul Independentei, 050095 Bucharest, Romania; simona.stroescu@inflpr.ro (S.N.); madalina-andreea.badea@bio.unibuc.ro (M.A.B.); anca.dinischiotu@bio.unibuc.ro (A.D.); 2Laser Department, National Institute of Laser, Plasma, and Radiation Physics, 409 Atomistilor Str., 077125 Magurele, Romania; mihai.boni@inflpr.ro (M.B.); andra.dinache@inflpr.ro (A.D.); adriana.smarandache@inflpr.ro (A.S.); ana.udrea@inflpr.ro (A.-M.U.); petronela.garoi@inflpr.ro (P.P.); 3Research Institute of the University of Bucharest (ICUB), University of Bucharest, 90-92 Sos. Panduri, 050663 Bucharest, Romania

**Keywords:** PDT, porphyrins, TMPyP4, TiO_2_ nanoparticles, melanoma, reactive oxygen species

## Abstract

The combination of TiO_2_ nanoparticles (NPs) and photosensitizers (PS) may offer significant advantages in photodynamic therapy (PDT) of melanoma, such as improved cell penetration, enhanced ROS production, and cancer selectivity. In this study, we aimed to investigate the photodynamic effect of 5,10,15,20-(Tetra-N-methyl-4-pyridyl)porphyrin tetratosylate (TMPyP4) complexes with TiO_2_ NPs on human cutaneous melanoma cells by irradiation with 1 mW/cm^2^ blue light. The porphyrin conjugation with the NPs was analyzed by absorption and FTIR spectroscopy. The morphological characterization of the complexes was performed by Scanning Electron Microscopy and Dynamic Light Scattering. The singlet oxygen generation was analyzed by phosphorescence at 1270 nm. Our predictions indicated that the non-irradiated investigated porphyrin has a low degree of toxicity. The photodynamic activity of the TMPyP4/TiO_2_ complex was assessed on the human melanoma Mel-Juso cell line and non-tumor skin CCD-1070Sk cell line treated with various concentrations of the PS and subjected to dark conditions and visible light-irradiation. The tested complexes of TiO_2_ NPs with TMPyP4 presented cytotoxicity only after activation by blue light (405 nm) mediated by the intracellular production of ROS in a dose-dependent manner. The photodynamic effect observed in this evaluation was higher in melanoma cells than the effect observed in the non-tumor cell line, demonstrating a promising potential for cancer-selectivity in PDT of melanoma.

## 1. Introduction

Skin diseases are spread all over the world, and the incidence of skin neoplasms is expected to increase in the next 20 years, especially in the Caucasian population [1]. Cutaneous melanoma results from a genetic and epigenetic transformation of melanocytes and accounts for only 10% of skin cancers, but about 80% of death are provoked by these [2].

The current strategies for melanoma treatment are surgery, radiotherapy, chemotherapy, immunotherapy, targeted therapy as well as photodynamic therapy [3].

Photodynamic therapy (PDT) is an anticancer strategy developed in the last century for the treatment of several types of cancer. [4]. The research for the use of PDT as a potential treatment option for melanoma was started in 1988 with two significant studies [5,6] that laid the foundation for the subsequent investigation of PDT for this type of cancer [7]. PDT consists in a two-stage therapy based on the use of light, oxygen, and a photosensitive molecule called photosensitizer (PS) which is non-toxic itself but in combination with the other two agents, triggers a photochemical reaction that leads up to tumor cell death. In the first stage of the PDT mechanism, PS is internalized by cells of the target tissue. In the second stage, PS is activated from the ground to an excited state under exposure to a specific wavelength of light. When it returns to its ground state, PS releases energy and transfers it from light to molecular oxygen, leading to the generation of singlet oxygen, a type of reactive oxygen species (ROS). When ROS amount overwhelms the cellular enzymatic and non-enzymatic antioxidant systems, oxidative stress occurs. ROS are extremely reactive and can attack biomolecules such as polyunsaturated fatty acids of phospholipids, DNA, and proteins. Depending on the magnitude of oxidative stress, necrosis (high level) or programmed cellular death (lower level), i.e., apoptosis and/or autophagy could be registered [8].

Compared with other therapeutic alternatives, PDT presents the advantages of limited damage to healthy cells, minimal invasiveness and systemic toxicity, limited resistance, and reduced adverse effects [9,10]. Even though PDT has key advantages for melanoma therapy, its efficiency is reduced due to the limitations brought by its components and mechanism [11].

Over the last years, blue light (400–500 nm) has been attracting more attention in PDT due to its higher energy compared to red or infrared light (760–1000 nm), which is beneficial in activating the PS used in PDT. Yet, it can only penetrate a relatively shallow depth into the tissue, which limits its use to superficial tumors or lesions that are located close to the surface of the skin. Blue light is primarily absorbed by the epidermis of the skin and can reach further into the dermis, up to the depth of 0.7–1 mm. The effect of blue light in PDT can vary depending on several factors, including the wavelength of the light, the absorption properties of the tissue, and the concentration of the PS used [12]. Overall, while the limited penetration depth of blue light is an important consideration in the use of PDT, by carefully selecting the appropriate PS and delivery method, blue light can be a valuable tool in the treatment of superficial tumors and lesions.

PDT of melanoma is also limited by resistance mechanisms driven mainly by elevated levels of melanin. These include optical interference by the highly-pigmented melanin, the anti-oxidant effect of melanin, the sequestration of PS inside melanosomes, defects in apoptotic pathways, and the efflux of PS by ATP-binding cassette (ABC) transporters [13]. Therefore, depigmentation or suppression of melanogenesis (tyrosinase activity) have been used as strategies to address the problems linked to melanin interference in PDT [14].

Nanoparticles (NPs) have shown the potential to improve PDT efficacy. For instance, NPs may deliver and concentrate PS in the cytoplasm of cancer cells and increase PS toxicity by providing a source of supplementary ROS. [15].

Due to their properties, TiO_2_ NPs may represent a promising candidate for PDT [16]. However, their use in this application is constrained by low tissue penetration of ultraviolet light and the harmful effects of ultraviolet radiation. To increase ROS formation and to improve the physicochemical properties, particularly the absorption of visible light, these NPs are conjugated with PS, such as porphyrins [17,18]. By electron transfer from the PS excited by visible light to the TiO_2_ NPs, ROS can be produced. In this way, an increase in photodynamic activity is achieved by the synergistic production of ROS by both PS and TiO_2_ [19]. This may help elevate the oxidative stress in melanoma cells, considering that melanin may act as a ROS scavenger contributing to resistance [20]. More, under UV irradiation, TiO_2_ NPs have been proven to be effective in preventing drug efflux caused by multidrug resistance (MDR), which enables enhanced drug accumulation in cells [21]. Others have indicated that TiO_2_ NPs increase the selectivity of porphyrin-based PS and reduce their adverse effects [16].

The goal of this study was to investigate the photodynamic effect of 5,10,15,20-(Tetra-N-methyl-4-pyridyl)porphyrin tetratosylate (TMPyP4) complexes with TiO_2_ NPs, on human melanoma cells by irradiation with 1 mW/cm^2^ blue light emitted by a 405 nm LED. UV-Vis absorption spectroscopy, Fourier transform infrared spectroscopy (FTIR), scanning electron microscopy (SEM), dynamic light scattering (DLS), and time-resolved phosphorescence of generated singlet oxygen was used to characterize TMPyP4/TiO_2_ complexes.

The toxicity profile and the water solubility of the porphyrin derivative were predicted using two web services, pkCSM [22] and ProTox-II [23]. PkCSM uses a new method that employs graph-based signatures to estimate a small molecule’s pharmacokinetics [22]. ProTox-II is a web tool that uses machine learning, frequent features, fragment propensities, and similarity to predict the toxicity of small molecules [23].

To explore the possible anticancer photoactivity of the porphyrin TMPyP4/TiO_2_ complex, the metabolic activity and the reactive oxygen species (ROS) and nitric oxide (NO) production were measured in human amelanotic melanoma cells in comparison to normal skin cells.

## 2. Materials and Methods

### 2.1. Chemical Compounds

The tested PS, 5,10,15,20-(Tetra-N-methyl-4-pyridyl)porphyrin tetratosylate (TMPyP4) (purity > 95%), was purchased from PorphyChem SAS (Dijon, France). The TiO_2_ NPs as anatase, with 99.5% purity and 17 nm diameter, were acquired from Nanografi (Ankara, Turkey).

The TiO_2_ NPs suspension was prepared in 20 mL of distilled water, 5 mg TiO_2_ was added into it by ultrasonication for 1 h at room temperature to obtain 0.25 mg/mL TiO_2_ suspension. For the TMPyP4 solution, 40 mg TMPyP4 was dissolved into 10 mL distilled water to obtain 4 mg/mL concentration; the solution was ultrasonicated for 1 h at room temperature. In the next step, 500 µL TMPyP4 (4 mg/mL) was added to the NP’s suspension and ultrasonicated for 2 h. After overnight incubation in the dark, the mixture was centrifuged at 10.000 rpm for 12 min. The supernatant was removed, and the precipitate was resuspended in distilled water. The entire protocol was performed according to the literature [24,25,26].

### 2.2. Morphological Characterization

A high-resolution scanning electron microscope (HRSEM) was used for topographic analysis of TiO_2_ and TMPyP4/TiO_2_ samples as well as to observe the structural quality and surface morphology of samples. More precisely, the Apreo microscope from FEI (Thermo Fisher Scientific, Waltham, MA, USA) allows a 0.9 nm resolution. By using HRSEM, we studied the surface of the samples at different magnifications (from 50,000× to 250,000×), in a high vacuum, by scanning them with a beam of accelerated electrons at very high energies (≈25 keV) at 10 cm working distance and an electrical current of 25 pA. Prior to imaging analysis, a thin layer of gold was sputtered onto the sample surfaces to avoid electrostatic charging during measurements.

The system is equipped with an X-ray source and an EDX unit with elementary energy dispersion spectroscopy (EDS), a fixed silicon detector, and an integrated Peltier element as a cooling system. For EDS, it was operated at 10 kV acceleration voltage, 6.3 pA electrical current, and the dead time was 40 s.

DLS measurements were recorded with Nanoparticle Analyzer SZ-100V2 (Horiba, Kyoto, Japan, which employs a diode-pumped solid-state laser emitting at 532 nm. The analyses were carried out in triplicate at 25 °C. The mean hydrodynamic size was measured at a scattering angle of 90° for a sample volume of 1 mL, placed in a quartz cuvette with 1 cm optical path length. The Zeta potential of the samples was recorded in an electrophoretic cell with carbon electrodes (6 mm).

### 2.3. Spectroscopic Methods

The TMPyP4 concentration in the TMPyP4/TiO_2_ suspensions was determined by UV Vis absorption spectroscopy in quartz cells of 1 cm path length, in the spectral range 200–800 nm. The measurements were achieved by using a spectrophotometer Lambda 950 (Perkin-Elmer, Waltham, MA, USA).

FTIR spectrometer (Nicolet iS50, Thermo Scientific, Waltham, MA, USA) in absorption mode with a resolution of 4 cm^−1^ was used to record the IR spectra. Each sample was placed on a KRS-5 crystal and dried in direct air flux.

The generation of singlet oxygen by the PS samples was investigated by time-resolved phosphorescence emitted at 1270 nm [27,28]. The excitation was made with the second harmonic generation (SHG) of a pulsed Nd:YAG laser (Minilite II, Continuum, Excel Technology, Milpitas, CA, USA), λ = 532 nm, 6 ns FWHM, 10 Hz, and the pulse energy 3 mJ. The solutions were placed in a quartz cuvette with an optical path of 1 cm. A near-IR photomultiplier tube (PMT) module (Hamamatsu H10330 Module) was placed at a 90-degree angle with respect to the incident laser beam to collect the signal. Interferential optical filters were used to eliminate wavelengths other than 1270. The PMT was coupled with a Digital Phosphor Oscilloscope (Tektronix type DPO-7254) to measure the signal.

### 2.4. Toxicity and Water Solubility Predictions

For the toxicity predictions, the TMPyP4 structure was imported as SMILES from the PubChem database (Figure 1) [29]. The toxicity prediction was made using two web services pkCSM [22] and ProTox-II [23]. The water solubility prediction has been made using the pkCSM web service. Those predictions required the SMILES code previously obtained.

### 2.5. Irradiation System

A custom system for cell culture plate irradiation was used for PDT experiments. This system was described in detail in [31]. The light source used was a 405 nm LED (Nikia, Japan, NVSU 233B). A Plano-Convex Lens, with a focal length of f = 100 mm, was situated 25 mm away from the light source to collect all radiation emitted by the diode. This was also positioned 30 mm from the well of the 96-well culture plates to irradiate the entire area of interest. The irradiance was 1 mW/cm^2^.

### 2.6. Cell Lines and Culture Conditions

Human cutaneous amelanotic melanoma cell line, Mel-Juso, (ACC-74, ATCC, Manassas, VA, USA), and normal skin fibroblasts from human CCD-1070Sk cell line (CRL-2091, ATCC, Manassas, VA, USA) were grown in RPMI 1640 medium (A1049-01, Gibco, Dublin, Ireland) and respectively MEM medium (61100-087, Gibco, Dublin, Ireland) both supplemented with 1% antibiotics-antimycotics solution (A5955, Sigma-Aldrich, St. Louis, MO, USA) and 10% fetal bovine serum (10270-106, FBS, Gibco, by Life Technologies, Carlsbad, CA, USA). Cell cultures were incubated in standard conditions (37 °C, 95% humidity, and 5% CO_2_). The culture media was completely refreshed once every two-three days.

### 2.7. Cell Treatment and Light-Irradiation Procedure

Mel-Juso and CCD-1070Sk cells were seeded in 96-well plates at a density of 10^4^ cells/well and 7.5 × 10^3^ cells/well, respectively. Following the overnight incubation, the cells were treated with different concentrations of TMPyP4/TiO_2_ complex, free TMPyP4 (5,10,15,20-(Tetra_N-methyl-4-pyridyl porphyrin tetratosylate), and TiO_2_ NPs. Before treatment, all suspensions were sterilized by UVC radiation for 30 min (no changes of absorption spectra detected) and then diluted in the cell culture media to the following concentrations: free TMPyP4—0.1, 0.25, 0.5, 0.75, 1 μg/mL and TiO_2_ NPs—4, 10, 20, 30.7, 40 μg/mL. Identical concentrations were used for the complex. Untreated cells served as control.

Cell cultures were incubated for 24 h to allow the internalization of TMPyP4 and NPs, and afterward, the culture medium was replaced with a fresh one. Immediately, the cells were light-irradiated for 7.5 min and incubated at 37 °C for 24 h. Non-irradiated cells (kept in the dark) were treated in similar conditions.

### 2.8. MTT Assay

Cell viability was assessed by colorimetric MTT assay using the MTT reagent (3-(4,5-dimethyl-2-thiazolyl)-2,5-diphenyl-2H-tetrazolium bromide, M5655, Sigma-Aldrich, St. Louis, MO, USA). After cell exposure to experimental conditions, the culture media from each well was removed and replaced with 80 μL of 1 mg/mL MTT solution. Further, the 96-well culture plates were incubated for 2 h at 37 °C. A volume of 150 μL isopropanol (33539, Sigma-Aldrich, St. Louis, MO, USA) was added to each well to solubilize the purple formazan crystals formed after the reduction in the yellow tetrazolium salt by metabolically active cells. The absorbance was measured at 595 nm using a Flex Station 3 microplate reader (Molecular Devices, San Jose, CA, USA).

### 2.9. Live/Dead Assay

The LIVE/DEAD Viability/Cytotoxicity Kit (L3224, Invitrogen, Gloucester, UK) was used to discriminate between live and dead cells based on esterase activity and plasma membrane integrity. After exposure to experimental conditions, the culture medium was removed, and the cells were treated with a serum-free solution containing 2.5 μM calcein AM and 4 μM ethidium homodimer. After 20 min of incubation at room temperature, the cells were observed by an inverted phase contrast Olympus IX73 microscope (Olympus, Tokyo, Japan) and photographed using CellSens Dimension software and FITC and TRITC filters.

### 2.10. Measurement of ROS Production by H_2_DCFDA Staining

The level of intracellular reactive oxygen species (ROS) was fluorometrically detected using the permeant 2′,7′-dichlorodihydrofluorescein diacetate compound (H_2_DCFDA, D6883, Sigma-Aldrich, St. Louis, MO, USA). This assay measures hydroxyl, peroxyl, and other ROS activity within the cell. Thus, after 24 h of treatment with TMPyP4, TiO_2_ NPs, and TMPyP4/TiO_2_ complex, the culture medium was removed, and the cells were incubated for 30 min at 37 °C, with 50 μM H_2_DCFDA solution prepared in HBSS (Hanks’ Balanced Salt Solution). Further, the H_2_DCFDA solution was replaced with a culture medium, and the cells were irradiated for 7.5 min. The fluorescence of product—dichlorofluorescein (DCF) was read at Flex Station 3 microplate reader (ex. 485 nm/em. 520 nm) after 24 h of incubation. The values of relative fluorescence units (RFU) were related to the total amount of protein estimated by the Bradford method in each well.

### 2.11. Quantification of NO Release by Griess Method

The Griess method [] [32] was used to estimate the production of nitric oxide (NO) from cells exposed to experimental conditions by measuring the nitric oxide-derived nitrite concentration from culture media. A volume of 80 μL media from each well was mixed with 80 μL Griess reagent containing sulfanilamide (S9251, Sigma-Aldrich) and N-(1-Naphthyl)ethylenediamine dihydrochloride (222488, Sigma-Aldrich) solutions (1:1). The absorbance of the samples was measured at 540 nm. The NO concentration of the samples was determined using a NaNO_2_ standard curve (0–100 μM).

### 2.12. Statistical Analysis

All determinations were performed in triplicate. The data were expressed as relative values in comparison with the control (100%) and calculated as mean ± standard deviation. The results were statistically analyzed in GraphPad Prism (Version 8, GraphPad Software, La Jolla, CA, USA), using the two-way ANOVA method and Dunnett’s multiple comparisons tests (treated cells vs. control). The values *p* < 0.05 (*), *p* < 0.01 (**), and *p* < 0.001 (***) were considered significant.

## 3. Results

### 3.1. Morphological Characterization of the TMPyP4/TiO_2_ Complex

The TiO_2_ sample is depicted in Figure 2A–D at various magnifications (50,000×, 100,000×, 150,000×, 250,000×). Presented SEM images were considered for the entire surface of the sample, and 6 to 8 different microscopic fields were selected for examination.

On the TiO_2_ sample, it can be observed from the SEM analyses the irregular spherical shaped particles with distinct structures (Figure 2A–D). These analyses employ different magnifications depending on the quality of the sample deposition layer and the structure of their surface.

The TMPyP4/TiO_2_ sample at different magnifications (50,000×, 100,000×, 150,000×, 250,000×) (see Figure 3A–D) show microspheres on the surface similar to the TiO_2_ sample (Figure 2A). However, the latter case (the SEM images corresponding to TMPyP4/TiO_2_ sample) has the microspheres adherent in an isolated form. Conglomerates were formed by the accumulation of TMPyP4/TiO_2_ NPs. These SEM images are in concordance with those reported in the literature [26,33] as well.

By SEM image analysis, it is observed that the particles are dispersed on a nanometer scale. In Figure 2D, the TiO_2_ synthesized sample shows a surface with NPs with dimensions between 9.4–19.5 nm (Table 1), but for TMPyP4/TiO_2_ sample (Figure 3D), the size distribution is between 25.4–33.6 nm. The microscopy results demonstrate that the synthesized TMPyP4/TiO_2_ complexes have the dimensions of the particles increased.

The TMPyP4/TiO_2_ sample exhibits a higher aggregation of grains (resulting in a more amorphous structure of coatings) than the much-ordered arrangement in the TiO_2_ sample.

The EDS analyses were conducted in duplicate on three distinct and non-overlapping regions having areas of 250 µm × 250 µm.

The results of EDS investigations for TiO_2_ and TMPyP4/TiO_2_ samples, Weight %—mass percentage, and Atomic %—atomic percentage are presented in Table 1. The analysis of the chemical composition of TiO_2_ samples showed mostly Ti content, O, and C, in addition to N and S atoms for TMPyP4/TiO_2_, without other alloying elements.

The topographical results revealed that the porphyrin was uniformly dispersed on the surface of the TiO_2_, and it is clear that the synthesis process has an important role in the quality of TMPyP4/TiO_2_ samples.

DLS technique provided information about the mean hydrodynamic size of TiO_2_ and TMPyP4/TiO_2_ suspensions, as well as values of the zeta potentials of the same samples. The recorded results are shown in Table 2.

The mean hydrodynamic size of the TiO_2_ NPs measured in suspension was 291.2 nm. This value suggests that the NPs form aggregates in the measured samples. For the TMPyP4/TiO_2_ complex, the mean hydrodynamic size was 823.3 nm, showing not only the formation of the complexes but also the aggregation of these complexes in larger-scale structures. Similar aggregates, having sizes between 1148 nm and 1226 nm, were also observed for other TiO_2_-porphyrin complexes [26]. The appearance of these large-size conglomerates is mainly due to the coalescence and compactization that appear during centrifugation. Subsequent ultrasonication cannot totally overturn these phenomena [31]. DLS sizes could also be larger because the hydrodynamic size is approximated for theoretical spheres that surround the aggregates. Additionally, the solvation shell impacts the size of the theoretical spheres measured through DLS [34]. The values of the polydispersity index, given in Table 2, indicate a broad distribution of the sizes of aggregates in both samples [35].

More so, Table 2 presents the values of the zeta potential for TiO_2_ NPs, as well as for the TMPyP4/TiO_2_ complex. The value −58.6 mV indicates good stability of the TiO_2_ suspensions. Zeta potential measured for the TMPyP4/TiO_2_ complex (−48.5 mV) suggests a slightly diminished stability in comparison to the TiO_2_ suspensions. Nevertheless, remaining in the interval characterizes good stability. Overall, a zeta potential larger than −30 mV (in absolute value) indicates that a suspension could be regarded for drug delivery [36].

### 3.2. Spectroscopic Analysis

Figure 4 exhibits the absorption spectrum for 0.01 mg/mL TMPyP4 solution in distilled water compared with the absorption spectra for TMPyP4/TiO_2_ complexes and 0.12 mg/mL TiO_2_ NPs water suspensions. The TMPyP4 spectrum shows absorption maxima corresponding to Soret and, respectively, Q bands at 422 nm and 518 nm, 560 nm, and 583 nm. The absorption bands of TMPyP4 are clearly seen in the spectrum of TMPyP4/TiO_2_ suspension in water and prove the functionalization of NPs with the PS.

The intensity of the 422 nm absorption peak of TMPyP4 was used to quantify the loading of porphyrin in the TMPyP4/TiO_2_ complexes. The inset of Figure 4 shows the calibration curve consisting of the absorbance values for the peak at 422 nm versus concentration for several TMPyP4 solutions in water with concentrations in the range of 1–10 µg/mL. Taking into account the 0.47 absorbance for TMPyP4/TiO_2_ suspension (Figure 4) at 422 nm (the background due to NPs diffusion being subtracted), we can extrapolate from the calibration curve a TMPyP loading of 3 µg/mL in the complexes. Similar calibration curves were used for TiO_2_ NPs by considering the absorbance of TiO_2_ suspension for 0.12 mg/mL (Figure 4) and that of TMPyP4/TiO_2_ at 700 nm (where no features of TMPyP4 are present), we can deduce a 0.1 mg/mL concentration of TiO_2_ in the complexes.

The FTIR spectra of TMPyP4, TiO_2_ NPs, and TMPyP4/TiO_2_ complexes were recorded and are shown in Figure 5.

The FTIR spectrum of TMPyP4 exhibits typical bands corresponding to N-H stretching vibrations of the pyrrole rings at 3320 cm^−1^ and C-H stretching vibrations at 3055, 2917, and 2850 cm^−1^ [37,38]. The rise of the band at 1640 cm^−1^ can be due to stretching vibrations of the C=N bonds in the pyridyl ring [39]. The C=C stretching combined with C-H and N-H bending vibrations can be assigned to the peaks raised between 1600–1200 cm^−1^ [26,40]. Signals related to SO_2_ stretching vibrations were detected at 1189 cm^−1^ [38]. In-plane scissoring vibrations of C-H bonds and N-H wagging appear at 1125, 1035 cm^−1^, and 970 cm^−1^. C-H deformations from the porphyrin macrocycle can be responsible for the peaks in the 900–700 cm^−1^ spectral range [31,38]. The N-H wagging vibrations coupled with out-of-plane bending vibrations of C-S bonds give rise to the peak at 683 cm^−1^ [31]. SO_2_ twisting and wagging vibrations can be assigned to bands below 555 cm^−1^ [37].

For bare TiO_2_, two broad absorption bands located between 3500–3000 cm^−1^ and around 1630 cm^−1^ correspond to the stretching and bending vibration of the hydroxyl group, showing the existence of water adsorption on the surface of TiO_2_ [41]. Besides, the band due to the Ti-O stretching vibration has been found in the range of 1000–400 cm^−1^ [37,42].

Compared with the simple TiO_2_, some characteristic bands of porphyrins are observed in the FTIR spectrum of TMPyP4/TiO_2_ samples which proves the loading of porphyrin on NPs. The increase in intensity for the peaks identified at 2917, 2850, 1574, and 1539 cm^−1^ that corresponds to the stretching vibration of C–H and C=C confirms the presence of TMPyP4 in the samples [42]. The interaction between the porphyrin ring core and TiO_2_ NPs is suggested by IR maxima emphasized at 1642 cm^−1^, which is specific to C=N in-plane vibrations, along with peaks centered at 1460, 1396, and 1348 cm^−1^, which can be assigned to distinct C=C in-plane vibrations from pyrroles. Likewise, the arising broad band centered at 1070 cm^−1^ associated with C-H in-plane deformation of pyrrole rings may be a hint of TMPyP4 attachment to TiO_2_ through the porphyrin ring core [43]. The broadband shaped between 1000–650 cm^−1^ (centered at 784 cm^−1^) can be associated with Ti–O–Ti vibrational modes interaction with N-H bending, C-H bending, C-C bending, C-N stretching vibrations as well as N-O and O-H deformation vibrations interactions. This may also be an indicator of TMPyP4/TiO_2_ complexes via N–O–Ti interactions between the pyrroles of the porphyrin ring and NPs [26]. The IR spectrum exhibits maxima at 465 and 437 cm^−1^, which can correspond to both Ti-O stretching vibrations and to out-of-plane bending of S-O bonds [37].

All these features bring evidence regarding the conjugation of TMPyP4 porphyrin derivative with TiO_2_ NPs.

Figure 6 shows the time-resolved phosphorescence signals measured at 1270 nm for the singlet oxygen generated by TMPyP4 water solution at 10 µg/mL and TMPyP4/TiO_2_ water suspension containing 1.5 µg/mL of TMPyP4. The signals were averaged over 1000 laser pulses and recorded in the same experimental arrangements for both samples.

The phosphorescence intensity is obtained by extrapolating at *t* = 0 the mono-exponential phosphorescence kinetics fitting curve. The value of intensity corresponding to the TMPyP4 solution is 0.0039 compared with 0.0016 for TMPyP4/TiO_2_. The concentration ratio for the two samples is 6.89, since for intensities, the ratio is 2.4. The enhancement of the signal in the case of TMPyP4/TiO_2_ suspensions can be assigned to scattering on TiO_2_ NPs.

### 3.3. TMPyP4 Toxicity Predictions

According to pkCSM and ProTox-II predictions, TMPyP4 has a RatLD50 of 2.482 mol/kg and 3066 mg/kg, respectively. TMPyP4 belongs to the low-toxicity class and is presumably neither mutagenic, immunotoxic, cytotoxic, nor carcinogenic.

TMPyP4 does not affect the AhR receptor, whose activation is associated with transcriptional changes linked to immunotoxicity, thymic involution, and immunosuppression [44]. Additionally, they are inactive on p53, ATAD5, and TMPyP4. According to the pkCSM and ProTox-II online sites, this porphyrin is not hepatotoxic. Overall, the predictions show that TMPyP4 is a non-toxic compound. In Table 3, we display the predicted results.

### 3.4. Dark and Light-Induced Cytotoxicity

The light-induced cytotoxicity of the synthesized TMPyP4/TiO_2_ complex was evaluated on human melanoma cells (Mel-Juso cell line), and normal skin fibroblast (CCD-1070Sk cell line) after irradiation with a 405 nm LED using MTT and Live/Dead assays. In parallel, cells treated with individual components of the complex at the same concentrations were analyzed in similar conditions. Untreated cells and treated cells exposed to dark conditions were also run as controls.

MTT assay studies revealed (Figure 7) that free TMPyP4 and TiO_2_ NPs are non-toxic in dark conditions or after 7.5 min light irradiation for both types of cells at the applied doses, with one exception. A significant decrease in cell viability by 13% compared to control was registered in the case of irradiated Mel-Juso cells treated with the highest dose of free TiO_2_ NPs.

The TMPyP4/TiO_2_ complex presented a dose-dependent cytotoxicity on both cell types (Figure 7C,F). However, a more pronounced decrease in cell viability was observed in the case of melanoma cells compared to normal skin fibroblast, especially under light-irradiation conditions. For control experiments, where cells were kept in the dark conditions, 63% of the Mel-Juso cells and 71% of the CCD-1070Sk cells survived at the highest concentration, while after 7.5 min light irradiation only 21% and 55%, respectively, of cells remained viable.

Phototoxicity of the synthesized complex on human melanoma and healthy skin cells was analyzed in more detail by Live/Dead staining. The highest three concentrations from each tested suspension were chosen after MTT evaluation. Fluorescence images (Figure 8) revealed a gradual increase in the number of dead cells (red fluorescence) with the increase in TMPyP4/TiO_2_ complex concentration only in the case of melanoma cells (Figure 8A). Interestingly, no dead cells or very few dead cells were noticed in the case of normal CCD-1070Sk fibroblasts. However, a decrease in live cell density in a concentration of TMPyP4/TiO_2_ complex-dependent manner was observed (Figure 8B). At the same doses, free TMPyP4 and TiO_2_ NPs presented no phototoxicity, with most cells exhibiting green fluorescence. This assay confirmed the results obtained by the MTT assay and revealed that the highest dose of PS induced an important cell death under light irradiation.

### 3.5. Effects of PDT on ROS and NO Production in Melanoma and Healthy Skin Cells

Studies on ROS production (Figure 9) revealed that synthesized TMPyP4/TiO_2_ complex subjected to 7.5 min light irradiation could generate ROS in a dose-dependent manner in skin cells (Figure 9F). In Mel-Juso cells, a significant overproduction of ROS (by 76% over control) was observed after treatment with TMPyP4/TiO_2_ complex, starting with a concentration of components of 0.5/20 μg/mL. At the same time, in the light-irradiated CCD-1070Sk skin cells, a significant amount of ROS (by 63% over control) was generated even at a lower concentration of complex (0.25/10 μg/mL). Despite this, at the highest dose of TMPyP4/TiO_2_ complex, the ROS level registered in Mel-Juso cells exceeded the one found in CCD-1070Sk cells by 51%. Free components of the complex exposed to light irradiation caused a limited production of ROS (Figure 9D,E). A significant elevation of ROS level compared to control was noticed in CCD-1070Sk cells treated with 1 μg/mL free TMPyP4 (by 46%) and with 20, 30.7, and 40 μg/mL TiO_2_ NPs alone (by 49%, 51%, and 35% respectively). In Mel-Juso cells treated with 40 μg/mL TiO_2_ NPs, the ROS level increased by 22% of the control. In dark conditions (Figure 9A–C), ROS were not generated in a significant manner in skin cells. A single statistically relevant (* *p* = 0.04) increase in ROS level (by 35%) was induced in CCD-1070Sk cells treated with the complex suspension having a concentration of components of 0.25/10 μg/mL.

No significant production of NO was detected under dark and light-irradiation conditions in treated Mel-Juso or CCD-1070Sk cells (Figure 10). NO was indirectly estimated by measuring the nitrite concentration from culture media after 24 h from the light-irradiation procedure. It is known that ROS production is closely linked to NO generation, and in certain conditions, elevated levels of ROS lead to low NO bioavailability [45].

## 4. Discussion

This study aimed to explore the potential activity of the TMPyP4/TiO_2_ complex in PDT against human amelanotic melanoma Mel-Juso cells in relation to the effects on normal CCD-1070Sk skin fibroblasts. Here, 5,10,15,20-Tetra-N-methyl-4-pyridyl porphyrin (TMPyP4) and TiO_2_ NPs were used as photosensitizers. We were interested in investigating the combined action of TMPyP4 and TiO_2_ NPs to increase the efficacy of PDT on melanoma, knowing that both are generators of ROS when activated by light. Thus, the efficiency of photosensitizers when synthesized as a complex was studied in comparison with their efficiency when used alone. The amelanotic melanoma cells used in this study represent a model of depigmented melanoma in which melanin interference is not involved.

TMPyP4 is a cationic porphyrin derivative that is considered a promising PS for PDT due to its properties, such as high solubility in water, high permeability through the cellular membrane, limited dark toxicity, and high selectivity for cancer cells over normal cells [46,47]. The mechanism of TMPyP4 phototoxicity involves the generation of reactive oxygen species (ROS) upon exposure to light resulting in the induction of cell death by apoptosis [48]. Specifically, when TMPyP4 absorbs light, it becomes excited and transfers an electron to molecular oxygen, creating a highly reactive form of oxygen known as singlet oxygen (^1^O_2_). This ^1^O_2_ can damage cellular components such as lipids, proteins, and DNA. For example, ^1^O_2_ can oxides guanine residue resulting in DNA fragments of different sizes. TMPyP4 possesses a high quantum yield of ^1^O_2_ (Φ = 0.77 ± 0.04) [44] and an absorption maximum in the wavelength range at 422 nm, which makes it suitable for surface tumors, skin wounds or sites accessible to an endoscope.

TiO_2_ NPs are prevailingly used in tissue and bone engineering due to their capacity to promote cell migration and adhesion [49,50], and these NPs serve as antibacterial agents through their ability to produce ROS in the presence of UV light [51,52]. Furthermore, they also can exhibit anticancer effects via intracellular ROS production [53]. Compared with iron oxide NPs (Fe_2_O_3_ NPs), which are mostly used as compounds-delivery systems for anticancer treatments by loading the NPs to target a specific binding protein or to control the nano-complexes with an external magnetic field [54,55], TiO_2_ NPs are not only a delivery vector; it also has a photocatalytic activity (generating ROS species) which moreover can be improved by functionalization with an appropriate sensitizer, such as porphyrins and phthalocyanines [19,56,57,58].

Currently, various types of NPs (zinc, cerium, iron, silver, and titanium oxide NPs) are promising candidates for biomedicine, and in the future, nanotechnology may become the most used method in clinical practice, with a considerable amount of research data in vitro and in vivo studies, due to the small size properties and large surface area, that give scientists new tools for identifying and treating diseases that were previously thought to be unapproachable.

TiO_2_ NPs demonstrated strong photodynamic activity [59] by generating ROS through redox reactions mediated by their photogenerated holes and electrons. The trapped electrons can act as reduction mediators and react with molecular oxygen to create superoxide anions (O_2_^●−^), while the surface-trapped positive holes react with donor molecules to produce hydroxyl radicals (HO^●^). These reactions can also undergo back reactions, where the holes and electrons can recombine with the reduced and oxidized species, respectively reducing the effectiveness of photocatalytic reactions. The quantity of ROS produced during these reactions depends on the morphology and size of the TiO_2_ NPs [60].

The morphological characterization of the synthesized TMPyP4/TiO_2_ complex revealed, through SEM images, that the formed complex has a nanometric size, and the porphyrin is uniformly distributed across the surface of TiO_2_ NPs.

The measured zeta potential of TiO_2_ NPs and TMPyP4/TiO_2_ showed that the samples have good stability, suggesting these suspensions could be suitable for medication delivery [36].

SEM analysis suggested aggregation of formed nanocomplexes. Although colloidal aggregation is generally regarded as an inconvenience during the drug discovery process, recent research has shown that colloidal small-molecule aggregates have therapeutic potential [61]. The generation of nanoscale aggregates is a modern approach to extending drug circulation time while preventing drug breakdown and unwanted harmful effects on the body whilst achieving the required therapeutic effect [62]. Colloidal small molecule aggregates can be used to inhibit not only specific enzymes [63] but also receptor proteins and protein-protein interactions [64]. Colloidal drug aggregates can also be used in a variety of delivery formulations [65]. Furthermore, nano-sized supramolecular aggregates have been proposed as a novel solution to the limitations of traditional molecular targets and therapeutics, particularly for in vivo use. Nano-sized supramolecular aggregates have interesting properties such as increased cell permeability, passive tumor targeting, easy surface engineering, and therapeutic payload delivery [66].

The physicochemical characteristics of the NPs, such as their size, shape, surface chemistry, architecture, and density, as well as the properties of the biological and biochemical environments, are parameters that influence the cellular uptake behavior [65]. Small agglomerates with sizes ranging from a few to several hundred nanometers can enter the cells via endocytosis-based mechanisms such as clathrin-mediated endocytosis, caveolae-mediated endocytosis, and micropinocytosis [67,68]. Internalization of TiO_2_-NPs occurs as well via endocytosis [69]. Thus, considering the size of the TMPyP4/TiO_2_ complex (823.3 nm), we tend to believe that it may enter through the same uptake mechanism. Once internalized, the TMPyP4/TiO_2_ complex could be distributed in the cytosol or trafficked to various subcellular compartments. Another possibility of internalization might be through membrane disruption. Previously it was shown that NPs could disrupt the cell membrane and enter the cell through the resulting membrane defects [70]. 3D reconstructed tomography also confirmed that TiO_2_ NPs aggregates were mainly distributed over the cell membrane surface [69].

The formation of the TMPyP4/TiO_2_ complex was evidenced through UV-Vis absorption spectroscopy, which allowed us to measure the concentration of TMPyP4 loaded on TiO_2_ NPs in the samples. Considering that at a concentration up to 10 µg/mL, no dimerization of TMPyP is observed (the calibration curves for TMPyP determination based on monomer-specific absorption peak at 422 nm showed a good linearity and, moreover, specific absorption bands of dimers are not present), we think that aggregation does not affect these measurements. Regarding NPs, similar aggregates are observed in the SEM images for TiO_2_ and TMPyP/TiO_2_, so, also, in this case, the absorption spectra should give a measure for the amount of TiO_2_ in the samples.

Time-resolved phosphorescence signal of singlet oxygen generated by the TMPyP4/TiO_2_ complex showed a higher value when compared with the signal of the TMPyP4 sample at a similar concentration. This can be due to the enhancement of the signal by scattering on NPs, and also, the TiO_2_ NPs can be photoactivated and yield ROS by electron transfer from the visible light excited porphyrin. [20].

In the present study, TMPyP4/TiO_2_ complex has been shown to cause a more significant light-induced cytotoxic effect on both Mel-Juso and CCD-1070Sk cells at all applied concentrations compared to the PS alone while the use of TMPyP4/TiO_2_ complex in dark conditions had a negligible effect on the cell viability of both cell lines. This study has also revealed good results in the reduction in Mel-Juso melanoma cell viability in comparison to CCD-1070Sk normal skin fibroblasts under irradiation for 7.5 min with a 405 nm light and 1 mW/cm^2^ power density proving the selectivity of the complex for cancer cells.

Previously it was reported that TMPyP4 alone induced no cytotoxicity to B78-H1 melanoma cells in dark conditions, but when it was irradiated with a halogen lamp at a fluence of 7.2 J/cm^2^, the metabolic activity of the cells started to decrease, at concentrations  <  0.15 μM (corresponding to 0.2 μg/mL) and their migration was inhibited [58]. However, in other cell types (A549, HeLa, U2OS, and SAOS2 cells), TMPyP4 favored cancer cell migration at low doses (0.25 or 0.5 μM) and inhibited cell proliferation at high doses (2 μM) after 72 h of exposure under non-irradiated condition [48].

Our results might suggest a significant contribution of TiO_2_ NPs to the phototoxicity of TMPyP4, resulting in a synergistic activity of the complex components.

Further, the results of the Live/Dead assay could indicate different phototoxic mechanisms in tumor versus normal cells, more specifically, induction of cell death in Mel-Juso melanoma cells and inhibition of cell proliferation of normal CCD-1070Sk skin fibroblasts. This also supports the hypothesis of the therapeutic use of the TMPyP4/TiO_2_ complex with the benefit of cancer cell selectivity. According to several previous studies, PDT promotes different types of cell death in melanoma cells dependent on various factors, including the dose of the PS, the energy of the applied light, the type of cells or PS, etc. For example, low doses of N-TiO_2_ NPs stimulate autophagy in human melanoma A375 cells, while high doses induce necroptosis [71]. Apoptosis [58] and necrosis [72] were also found to be activated by PDT in melanoma cells when porphyrin was used as PS.

We showed that the photodynamic activity of the TMPyP4/TiO_2_ complex in melanoma and skin fibroblasts relies on ROS generated after irradiation. In dark conditions, the level of ROS in both types of cells appears to be closer to the one registered in control cells. When we applied a 405 nm light irradiation for 7.5 min with 1 mW/cm^2^ power density, ROS were produced in a significant amount dependent on concentration, PS, and cell type. The level of ROS increased similarly in melanoma cells and normal fibroblasts exposed to TMPyP4/TiO_2_ complex. Enhanced ROS production observed in cells exposed to TMPyP4/TiO_2_ complex in light-irradiated conditions can be a consequence of the occurrence of a series of chemical reactions between the generated photoproducts of complex components, which may lead to the additional formation of more toxic species. The ROS generated by light-irradiated TMPyP4/TiO_2_ complex may include ^1^O_2_, trapped or free, HO^●^, and O_2_^●−^, as well as hydrogen peroxide (H_2_O_2_). H_2_O_2_ can be generated through the reaction between ^1^O_2_ and water, whereas HO^●^ can be generated by the reaction between H_2_O_2_ and metal ions such as copper or iron ones. H_2_O_2_ with HO^●^ can form protonated superoxide radical (HOO^●^), and the coupling of two HOO^●^ may result in the formation of H_2_O_2_ [73]. ROS are used in PDT to destroy cancer cells, but they can also have harmful side effects, such as inflammation or damage to healthy tissues [74].

Cancer cells and normal cells present differences in their redox homeostasis [75,76]. Cancer cells with an antioxidant system already triggered are more sensitive to enhanced intracellular ROS than normal cells and are less capable of achieving redox balance [77]. Therefore, by inducing ROS under these metabolic conditions, a high percentage of cancer cells undergo death.

Here, the CCD-1070Sk fibroblasts proved to be more sensitive to ROS induction compared to melanoma cells when high concentrations of PS were applied. However, because of their efficient antioxidant capacity, normal cells might resist better ROS attacks in order to survive. In contrast, cancer cells develop powerful antioxidant mechanisms, but an excess of ROS can interfere with the altered signaling pathways leading to severe cell damage and death [75]. This might explain the higher photodynamic efficiency of the TMPyP4/TiO_2_ complex on melanoma cells versus normal skin fibroblasts. Our data corresponds to previous studies showing the dependence of ROS production on PS concentrations and cell viability [78,79]. More, PDT could allow controlled photo-activated induction of ROS with different cellular outcomes. For instance, Moosavi et al. (2016) showed that visible-light irradiation of N-TiO_2_ NPs triggered a ROS-mediated autophagic response that could be fine-tuned to selectively induce differentiation or apoptosis in leukemia K562 cells [80].

ROS production can influence NO generation. For example, superoxide (O_2_^●−^) can react with NO to form peroxynitrite (ONOO^−^), a strong oxidant molecule that can cause cellular damage. Superoxide dismutase (SOD) serves as the primary defense against O_2_^●−^ mediated toxicity and also plays a role in cell signaling by regulating ROS levels (such as O_2_^●−^ and H_2_O_2_) while preserving the availability of NO [81]. In certain cases, high levels of ROS can decrease the production of NO, while in other cases, high levels of NO may be necessary to neutralize the oxidative stress induced by ROS [82].

Previous research has shown that NO plays several roles in the development of melanoma, including immune surveillance, apoptosis, angiogenesis, melanogenesis, and effects on the melanoma cells themselves. In general, high levels of NO are indicative of a negative prognosis for individuals with melanoma. In melanoma tumor cells, NO is generated via multiple mechanisms and can have both tumor-promoting and inhibitory effects. More, it was shown that in the presence of blue light, photolabile NO derivatives photodecompose to NO via a copper (Cu^1+^) dependent mechanism [83].

In our study, the measured NO levels in both cell lines were close to the levels of NO in the cells of the control group. Even at the application of the highest dose, which resulted in nearly 80% cell death of melanoma cells, the TMPyP4/TiO_2_ complex produced no significant change in the NO level. Moreover, none of the conditions used in this study caused NO release. Thus, it was considered that NO release did not have a significant role in the PDT mediated by the combination of TMPyP4 with TiO_2_ NPs on Mel-Juso and CCD-1070Sk cells.

## 5. Conclusions

This study showed the photodynamic efficacy of 5,10,15,20-(Tetra-N-methyl-4-pyridyl)porphyrin tetratosylate (TMPyP4) complexes with TiO_2_ NPs, on human cutaneous amelanotic melanoma cells by irradiation with 1 mW/cm^2^ blue light.

Morphological analyses of the samples determined the nanoscale dimensions of the formed TMPyP4/ TiO_2_ complex. FTIR spectrum of the TMPyP4/TiO_2_ complex showed not only vibrational bands characteristic to TiO_2_ and TMPyP4 but also bands specific to the formed complex, evidencing the loading of porphyrin on TiO_2_ NPs. UV-Vis absorption spectroscopy allowed us to determine the amount of TMPyP4 loaded on TiO_2_ NPs.

TMPyP4/TiO_2_ complex exhibited good efficiency in the generation of singlet oxygen.

The predictions indicated a low degree of toxicity for TMPyP4, suggesting that it is probably safe for utilization.

We have shown that in PDT, binding of the TMPyP4 with TiO_2_ NPs may enhance their efficiency against human melanoma Mel-Juso cells while being less phototoxic on normal CCD-1070Sk skin fibroblasts, encouraging the premise of the therapeutic use of TMPyP4/ TiO_2_ complex with the benefit of cancer cells selectivity. TMPyP4/TiO_2_ complex exhibits significant phototoxicity at doses ≥ 0.1/4 μg/mL in Mel-Juso cells and at doses ≥ 0.25/10 μg/mL in CCD-1070Sk cells as a result of an increased ROS production dependent on dose. The results indicate that the photodynamic effect of the TMPyP4/TiO_2_ complex may act by different mechanisms by inducing cell death in melanoma cells and inhibiting the proliferation of normal skin fibroblasts under visible light irradiation. However, no contribution of NO to the photodynamic activity of the TMPyP4/TiO_2_ complex was found. Further studies regarding the type of cell death as well as the signaling pathways involved in are necessary.

## Figures and Tables

**Figure 1 pharmaceutics-15-01194-f001:**
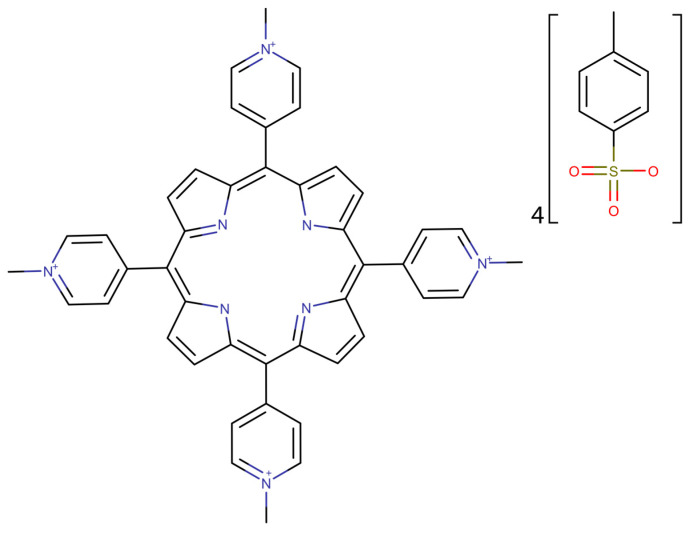
2D structure of TMPyP4 generated using the SMILES code imported from PubChem database [30].

**Figure 2 pharmaceutics-15-01194-f002:**
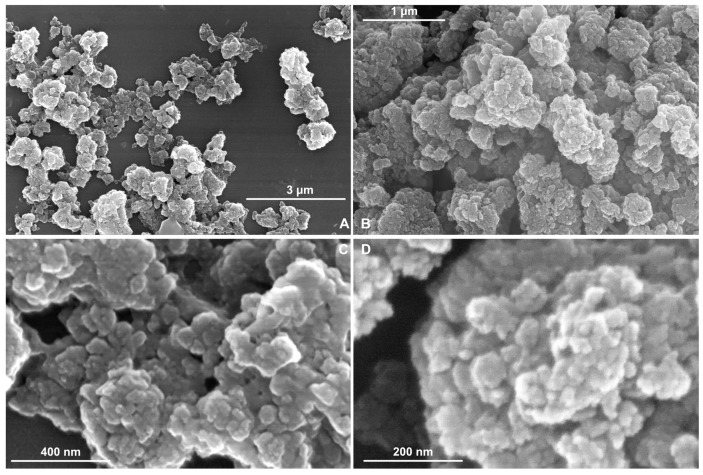
SEM images of TiO_2_ sample at different magnifications (**A**) 50,000*×*; (**B**) 100,000*×*; (**C**) 150,000*×*; (**D**) 250,000*×*.

**Figure 3 pharmaceutics-15-01194-f003:**
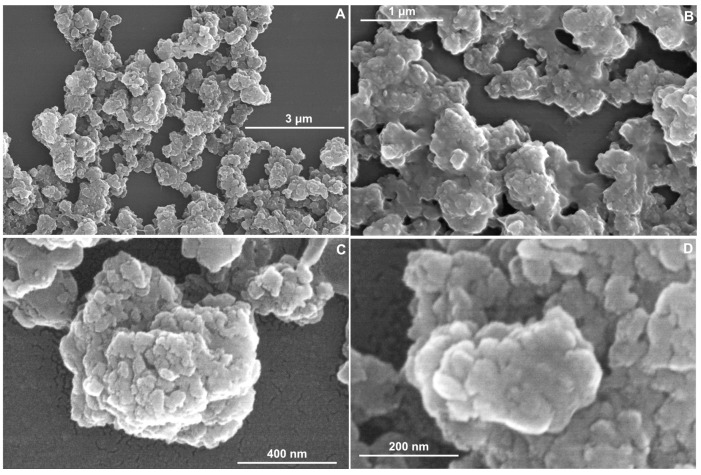
SEM images of TMPyP4/TiO_2_ sample at different magnifications (**A**) 50,000*×*; (**B**) 100,000*×*; (**C**) 150,000*×*; (**D**) 250,000*×*.

**Figure 4 pharmaceutics-15-01194-f004:**
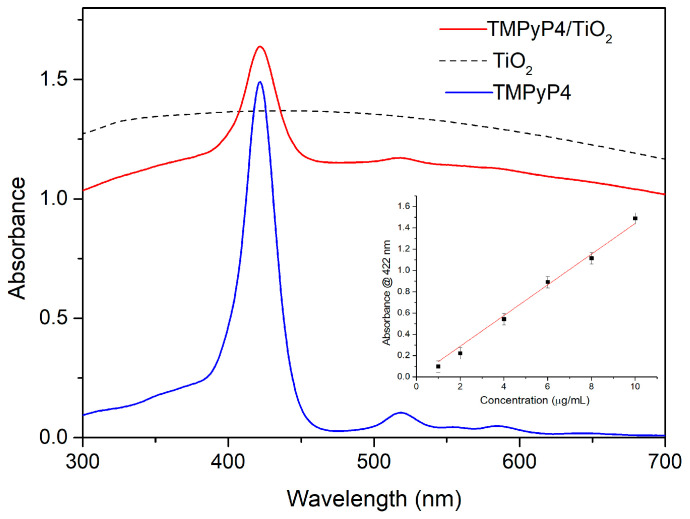
The absorption spectra for 0.01 mg/mL TMPyP4 water solution, 0.12 mg/mL TiO_2_ water suspension, and TMPyP4/TiO_2_ complexes water suspension. The inset shows the calibration curve for the determination of TMPyP4 concentration in the complexes.

**Figure 5 pharmaceutics-15-01194-f005:**
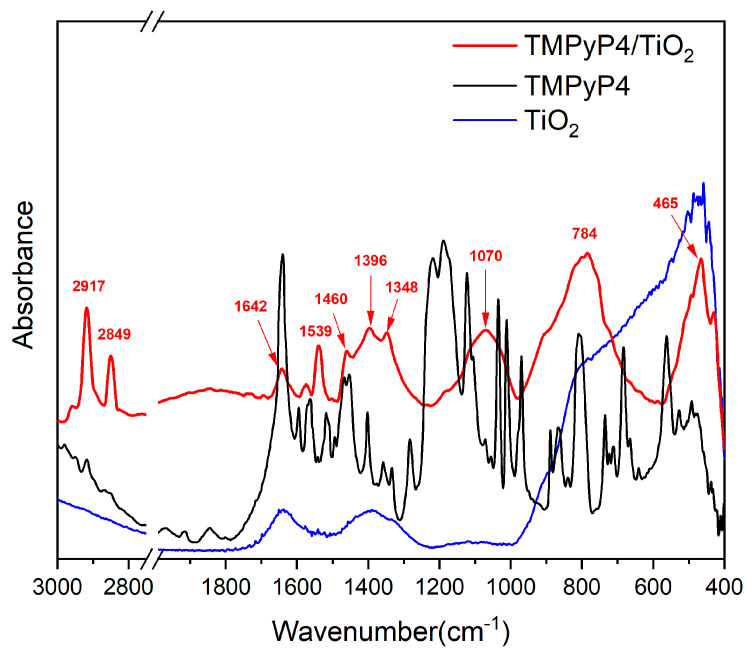
FTIR spectra of suspensions containing TiO_2_ nanoparticles, TMPyP4, and TMPyP4-loaded TiO_2_ nanoparticles.

**Figure 6 pharmaceutics-15-01194-f006:**
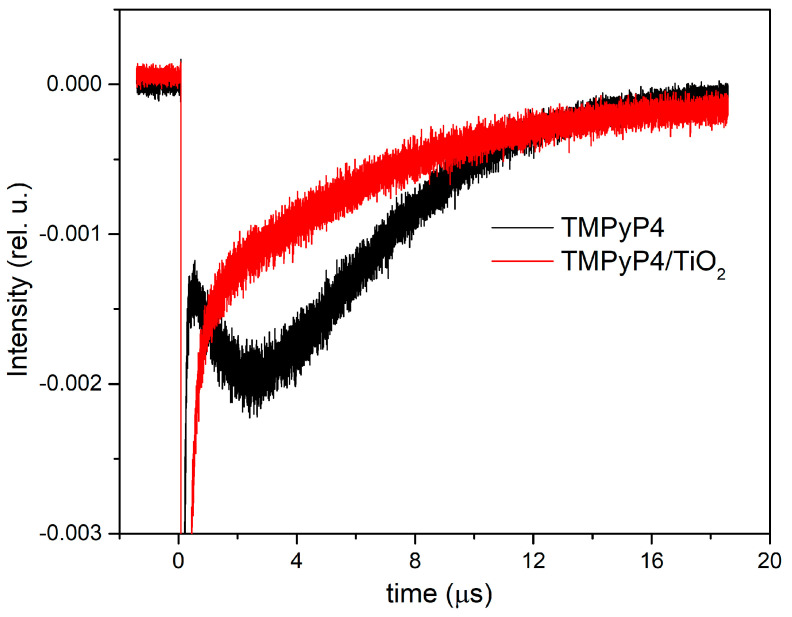
Time-resolved phosphorescence signals of singlet oxygen generated by the investigated samples.

**Figure 7 pharmaceutics-15-01194-f007:**
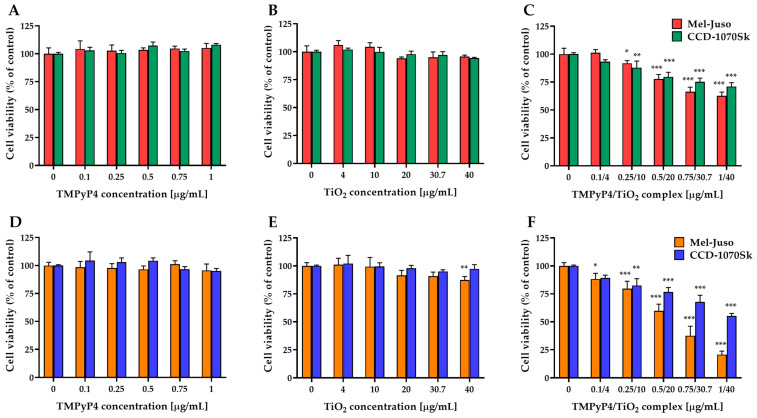
Effect of TMPyP4, TiO_2_ NPs, and TMPyP4/TiO_2_ complex on the metabolic activity/cell viability of cells. The graphs present the relative mitochondrial dehydrogenase activity of treated human Mel-Juso and CCD-1070Sk cells under dark (**A**–**C**) and light-irradiation conditions (**D**–**F**). Untreated cells (0 μg/mL) were used as a control. Results (control vs. sample) were significant at *p* < 0.05 (*), *p* < 0.01 (**), and *p* < 0.001 (***). Error bars reflect the standard deviation.

**Figure 8 pharmaceutics-15-01194-f008:**
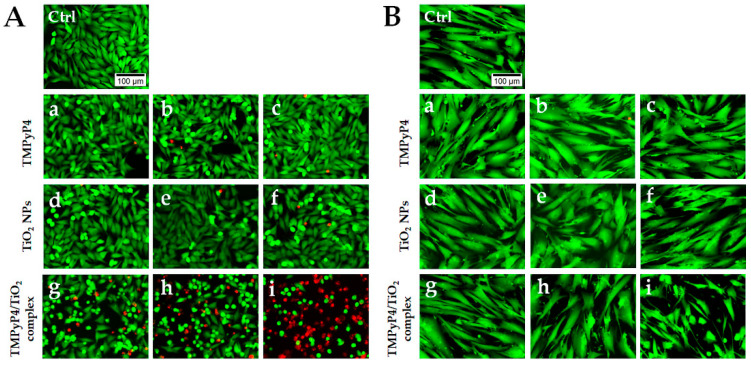
LIVE/DEAD staining on treated Mel-Juso (**A**) and CCD-1070Sk cells (**B**) after light irradiation for 7.5 min. Cells were treated with different concentrations of TMPyP4: 0.5 μg/mL (a), 0.75 μg/mL (b), and 1 μg/mL (c); TiO_2_ NPs: 20 μg/mL (d), 30.7 μg/mL (e) and 40 μg/mL (f) and TMPyP4/TiO_2_ complex: 0.5/20 μg/mL (g), 0.75/30.7 μg/mL (h), 1/40 μg/mL (i). Control cells (Ctrl) are non-treated, and light irradiated. Live cells are stained with calcein AM (green fluorescence), and dead cells with ethidium homodimer (red fluorescence). The scale bar is the same for all images and is 100 µm.

**Figure 9 pharmaceutics-15-01194-f009:**
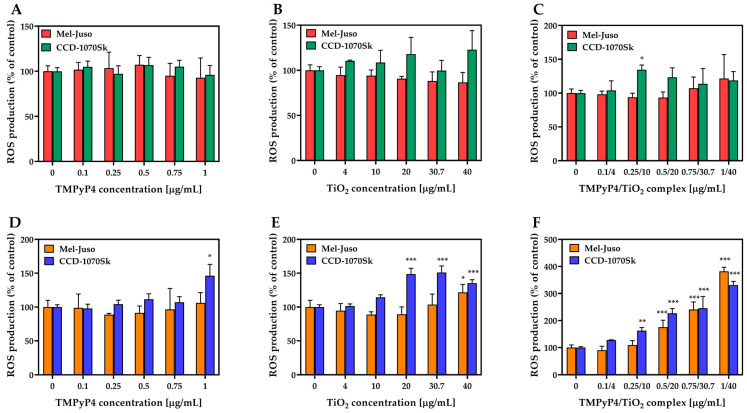
Effect of TMPyP4, TiO_2_ NPs, and TMPyP4/TiO_2_ complex on intracellular reactive oxygen species (ROS) production. The graphs present the relative level of dichlorofluorescein (DCF) fluorescence detected in treated human Mel-Juso and CCD-1070Sk cells under dark (**A**–**C**) and light-irradiation conditions (**D**–**F**). Untreated cells (0 μg/mL) were used as a control. Results (control vs. sample) were significant at *p* < 0.05 (*), *p* < 0.01 (**), and *p* < 0.001 (***). Error bars reflect the standard deviation.

**Figure 10 pharmaceutics-15-01194-f010:**
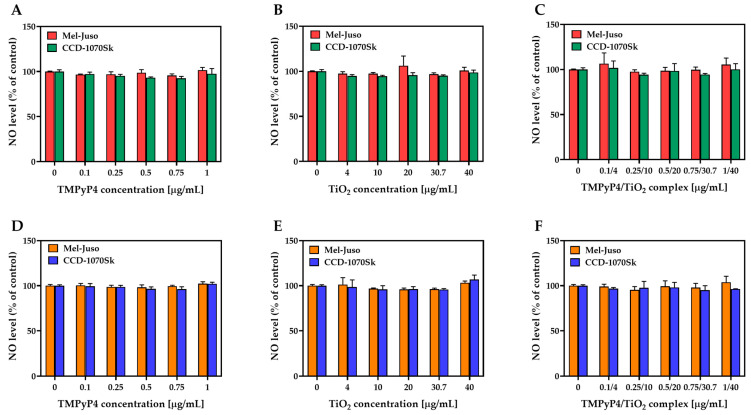
Effect of TMPyP4, TiO_2_ NPs, and TMPyP4/TiO_2_ complex on nitric oxide (NO) production. The graphs present the relative amount of nitrite released in culture media by treated human Mel-Juso and CCD-1070Sk cells under dark (**A**–**C**) and light-irradiation conditions (**D**–**F**). Untreated cells (0 μg/mL) were used as a control. Error bars reflect the standard deviation.

**Table 1 pharmaceutics-15-01194-t001:** Dimension of the particles and EDS analysis of the TiO_2_ and TMPyP4/TiO_2_ samples.

Sample	Size/nm SEM	Element	Weight %	Atomic %
TiO_2_	19	C K	3.54	4.62
		O K	35.34	60.52
		Ti K	61.12	34.86
TMPyP4/TiO_2_	28	C K	7.63	15.56
		N K	0.63	1.10
		O K	34.37	52.59
		S K	5.68	4.33
		Ti K	51.69	26.42

**Table 2 pharmaceutics-15-01194-t002:** The mean hydrodynamic size and zeta potential of the samples.

Sample	Mean Size (nm)	Standard Deviation (nm)	Polydispersity Index	Zeta Potential (mV)
TiO_2_	291.2	25.8	1.404	−58.6
TMPyP4/TiO_2_	823.3	148.9	0.880	−48.5

**Table 3 pharmaceutics-15-01194-t003:** Toxicity predictions for TMPyP4, using pkCSM and ProTox-II web services.

**pkCSM**		
**Model**	**Predicted Values**	**Unit**
Hepatotoxicity	No	Yes/No
AMES toxicity	No	Yes/No
hERG I inhibitor	No	Yes/No
hERG II inhibitor	No	Yes/No
Oral Rat Acute Toxicity (LD50)	2.482	mol/kg
**ProTox-II**		
**Model**	**Predicted values**	**Unit**
Predicted LD50	3066	mg/kg
Predicted Toxicity Class	5	1-bad 6-good
Model	Results	Probability
Hepatotoxicity	Inactive	0.67
Carcinogenicity	Inactive	0.60
Immunotoxicity	Inactive	0.88
Mutagenicity	Inactive	0.57
Cytotoxicity	Inactive	0.64
Aryl hydrocarbon Receptor (AhR)	Inactive	0.85
Androgen Receptor Ligand Binding Domain (AR-LBD)	Inactive	0.98
Estrogen Receptor Ligand Binding Domain (ER-LBD)	Inactive	0.96
Peroxisome Proliferator Activated Receptor Gamma (PPAR-Gamma)	Inactive	0.99
Nuclear factor (erythroid-derived 2)-like 2/antioxidant responsive element (nrf2/ARE)	Inactive	0.97
Phosphoprotein p53	Inactive	0.84
ATPase family AAA domain-containing protein 5 (ATAD5)	Inactive	0.96

## Data Availability

Not applicable.

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
