# Peer review of "Photodynamic Activity of TMPyP4/TiO2 Complex under Blue Light in Human Melanoma Cells: Potential for Cancer-Selective Therapy"

_pharmaceutics, 2023, doi:10.3390/pharmaceutics15041194_

Round 1

Reviewer 1 Report

In this work, the authors reported TMPyP4/TiO2 Complex for photodynamic therapy of melanoma cells. Some specific issues should be addressed before further consideration.

1. In Figure 2 and Figure 3, TiO2 and TMPyP4/TiO2 was seemed to be aggregated, which was not suitable for biomedical applications.

2. The particle size of TMPyP4/TiO2 was 823.3 nm. And cellular uptake behaviors of TMPyP4/TiO2 should be investigated.

3. Light toxicity of TMPyP4 should be evaluated to highlight the advantages of TMPyP4/TiO2.

4. Blue light was employed for photodynamic therapy, the limited light penetration depth should be discussed.

5. The mechanism of cancer selective therapy should be discussed.

Author Response

Review #1

Reply: We thank the reviewer for their careful reading of the manuscript and their constructive remarks. We have taken the comments on board to improve and clarify the manuscript. Please find below a detailed point-by-point response to all comments (reviewers’ comments in black, our replies in blue).

In this work, the authors reported TMPyP4/ TiO2 Complex for photodynamic therapy of melanoma cells. Some specific issues should be addressed before further consideration.

  1. In Figure 2 and Figure 3, TiO2 and TMPyP4/TiO2 was seemed to be aggregated, which was not suitable for biomedical applications.

Reply: We thank the reviewer for this observation. We take this opportunity to introduce a few comments in the Discussion Section (lines 621-633) referring to this aspect: “SEM analysis suggested aggregation of formed nanocomplexes. Although colloidal aggregation is generally regarded as an inconvenience during the drug discovery process, recent research has shown that colloidal small-molecule aggregates have therapeutic potential [64]. The generation of nanoscale aggregates is a modern approach to extending drug circulation time while preventing drug breakdown and unwanted harmful effects on the body whilst achieving the required therapeutic effect [65]. Colloidal small molecule aggregates can be used to inhibit not only specific enzymes [66], but also receptor proteins and protein-protein interactions [67]. Colloidal drug aggregates can also be used in a variety of delivery formulations [70]. Furthermore, nano-sized supramolecular aggregates have been proposed as a novel solution to the limitations of traditional molecular targets and therapeutics, particularly for in vivo use. Nano-sized supramolecular aggregates have interesting properties such as increased cell permeability, passive tumor targeting, easy surface engineering, and therapeutic payload delivery [68].”

  1. The particle size of TMPyP4/ TiO2 was 823.3 nm. And cellular uptake behaviors of TMPyP4/TiO2 should be investigated.

Reply: We thank the reviewer for this suggestion. We introduced the following paragraph in the Discussion Section (lines 634-647): “The physicochemical characteristics of the nanoparticle, such as its size, shape, surface chemistry, architecture, and density, as well as the properties of the biological and biochemical environments, are parameters that influence the cellular uptake behavior [69]. Small agglomerates with sizes ranging from a few to several hundred nanometers can enter the cells via endocytosis-based mechanisms such as clathrin-mediated endocytosis, caveolae-mediated endocytosis, and micropinocytosis [70,71]. Internalization of TiO2-NPs occurs as well via endocytosis [72]. Thus, considering the size of TMPyP4/TiO2 complex (823.3 nm) we tend to believe that it may enter through the same uptake mechanism. Once internalized, the TMPyP4/TiO2 complex could be distributed in the cytosol or trafficked to various subcellular compartments. Another possibility of internalization might be through membrane disruption. Previously it was shown that NPs can disrupt the cell membrane and enter the cell through the resulting membrane defects [73]. 3D reconstructed tomography also confirmed that TiO2 NPs aggregates were mainly distributed over the cell membrane surface [74].”

  1. Light toxicity of TMPyP4 should be evaluated to highlight the advantages of TMPyP4/TiO2.

Reply: We thank the reviewer for this comment but we fear the reviewer may have misunderstood us here. The light-induced toxicity of free TMPYP4 as well as of TIO2 NPs alone was already evaluated in parallel with TMPyP4/TiO2 complex. Please see figures 7 to 10. In the present study, we showed that in comparison to free TMPyP4, the TMPyP4/TiO2 complex presents higher toxicity after light irradiation thus highlighting the efficacy of the combination of the two components.

  1. Blue light was employed for photodynamic therapy, the limited light penetration depth should be discussed.

Reply: We thank the reviewer for this comment and the opportunity to improve our manuscript. A paragraph was added in the Introduction Section (lines 98-108): “Over the last years, blue light (400–500 nm) has been attracting more attention in PDT due to its higher energy compared to red or infrared light (760–1000 nm) which is beneficial in activating the PS used in PDT. Yet, it can only penetrate a relatively shallow depth into the tissue which limits its use to superficial tumors or lesions that are located close to the surface of the skin. Blue light is primarily absorbed by the epidermis of the skin and can reach further into the dermis, up to the depth of 0.7 - 1 mm. The effect of blue light in PDT can vary depending on several factors, including the wavelength of the light, the absorption properties of the tissue, and the concentration of the PS used [12]. Overall, while the limited penetration depth of blue light is an important consideration in the use of PDT, by carefully selecting the appropriate photosensitizer and delivery method, blue light can be a valuable tool in the treatment of superficial tumors and lesions.”

  1. The mechanism of cancer selective therapy should be discussed.

Reply: We have provided additional information to explain the mechanism of cancer selective therapy. The following sentences were included in the Discussion section (lines 735-739, 748-752): “Cancer cells and normal cells present differences in their redox homeostasis [79,80]. Cancer cells with an antioxidant system already triggered are more sensitive to enhanced intracellular ROS than normal cells and are less capable of achieving redox balance [81]. Therefore, by inducing ROS under these metabolic conditions, a high percentage of cancer cells undergo death.”

“More, PDT could allow controlled photo-activated induction of ROS with different cellular outcomes. For instance, Moosavi et al., (2016) showed that visible-light irradiation of N-TiO2 NPs triggered a ROS-mediated autophagic response that could be fine-tuned to selectively induce differentiation or apoptosis in leukemia K562 cells [84] ”.

Reviewer 2 Report

The research paper “Photodynamic Activity of TMPyP4/TiO2 Complex under Blue Light in Human Melanoma Cells: Potential for Cancer-Selective Therapy” by M.Ballas and A. Staicu describes synthesis and evaluation of TMPyP4/TiO2 nanoparticles for PDT of melanoma. Due to notoriously known aggressive behavior, melanoma is responsible for almost 80% of deaths among skin cancers, therefore further development of selective and efficient drugs is of high demand. In the paper by Ballas and Staicu, the authors proposed immobilization of photosensitizer (TMPyP4) on TiO2 NPs, which was meant to obtain novel drugs. The morphology and stability of NPs was studied be means of by SEM and dynamic light scattering (DLS). To prove that TMPyP4 was successfully immobilized on TiO2, the authors used UV-Vis absorption spectrophotometry as well as IR spectroscopy. In vitro dark toxicity and photodynamic activity of TMPyP4/TiO2 were assessed in human melanoma cells (Mel-Juso cell line) and normal skin fibroblast (CCD- 425 1070Sk cell line).

In my opinion, the paper by M.Ballas and A. Staicu is important for scientists seeking novel treatment options for melanoma. The authors demonstrated pronounced antiproliferative behavior for TMPyP4/TiO2 in human melanoma cells which was greater than that of observed effect in normal skin fibroblast.

However, several points should be answered before publication, my biggest concern is that the manuscript is incomprehensibly written:

1)      I strongly suggest reconceptualization the manuscript’s text.

a)      The abstract is too long.

b)     The introduction should be reorganized with a focus on how conjugation of PSs and TiO2 can help melanoma treatment. Clarifying figures would be useful. Please, use less well-known information and make the text shorter.

c)      Generally, melanoma is a type of cancer hard to treat by PDT due to elevated levels of melanin ( 10.3389/fonc.2020.00597 ). Please, discuss this point in the introduction. How developed NPs can or cannot overcome this resistance mechanism?

The statement “Photodynamic therapy (PDT) is an anticancer strategy developed in the last century for the treatment of melanoma…” is not true.

d)     The discussion is also too long and basically repeats the results and the introduction. Please, reorganize it and answer the question- “What is the mechanism behind the photodynamic activity enhancement when TiO2 NPs are conjugated with TMPyP4?”

Who absorbs the light (NPs or PSs) and what happens next? I assume, there is SET involved in the process, but I don’t clearly understand who generates ROS in the end.  

e)     In the discussion, please compare other NPs proposed by your group or other groups with novel NPs and give several statements on how you see further development of the field.

2)      “Taking into consideration the absorbance of 0.48 for the TMPyP4/TiO2 suspension at 422 nm (the background due to NPs diffusion subtracted), we can infer a concentration of 3 µg/mL for TMPyP4 in the TMPyP4/TiO2 complexes, corresponding to 90µg/ml of TiO2.”

Could you elaborate on how you measure these concentrations?

In the manuscript you mentioned that aggregates are formed in solutions. How this fact affects such measurements?

3)     It would be better to evaluate ROS generation yields rather than phosphorescence, since it would give quantitative results. If its possible, I would suggest doing that.

4)     I think, TMPyP4 can easily absorb fluorescence of H2DCFDA, is it an issue? How accurate are such studies (Figure 9)?

5)     The way I see it, toxicity and water solubility predictions are not well-justified. You have your NPs in hand. Why do you need to study them through computations?  TMPyP4 has been extensively studied in vivo (Mol Cancer Ther (2002) 1 (8): 565–573) and its toxicity profile is known. Also, I believe that solubility and toxicity of TMPyP4 will be different after immobilization on NPs. 

Author Response

Review #2

Reply: We thank the reviewer for their careful reading of the manuscript and their constructive remarks. We have taken the comments on board to improve and clarify the manuscript. Please find below a detailed point-by-point response to all comments (reviewers’ comments in black, our replies in blue).

The research paper “Photodynamic Activity of TMPyP4/TiO2 Complex under Blue Light in Human Melanoma Cells: Potential for Cancer-Selective Therapy” by M.Ballas and A. Staicu describes synthesis and evaluation of TMPyP4/TiO2 nanoparticles for PDT of melanoma. Due to notoriously known aggressive behavior, melanoma is responsible for almost 80% of deaths among skin cancers, therefore further development of selective and efficient drugs is of high demand. In the paper by Ballas and Staicu, the authors proposed immobilization of photosensitizer (TMPyP4) on TiO2 NPs, which was meant to obtain novel drugs. The morphology and stability of NPs was studied be means of by SEM and dynamic light scattering (DLS). To prove that TMPyP4 was successfully immobilized on TiO2, the authors used UV-Vis absorption spectrophotometry as well as IR spectroscopy. In vitro dark toxicity and photodynamic activity of TMPyP4/TiO2 were assessed in human melanoma cells (Mel-Juso cell line) and normal skin fibroblast (CCD- 425 1070Sk cell line).

In my opinion, the paper by M.Ballas and A. Staicu is important for scientists seeking novel treatment options for melanoma. The authors demonstrated pronounced antiproliferative behavior for TMPyP4/TiO2 in human melanoma cells which was greater than that of observed effect in normal skin fibroblast.

However, several points should be answered before publication, my biggest concern is that the manuscript is incomprehensibly written:

1)      I strongly suggest reconceptualization the manuscript’s text.

Reply: We agree with the reviewer’s suggestion and we thank him for this opportunity to improve our manuscript.

  1. a)      The abstract is too long.

Reply: Accordingly, to the reviewer’s comment, the abstract text was shortened.

  1. b)     The introduction should be reorganized with a focus on how conjugation of PSs and TiO2 can help melanoma treatment. Clarifying figures would be useful. Please, use less well-known information and make the text shorter.

Reply: Accordingly, to the reviewer’s comment, the introduction section was reorganized and shortened.

  1. c)      Generally, melanoma is a type of cancer hard to treat by PDT due to elevated levels of melanin ( 10.3389/fonc.2020.00597 ). Please, discuss this point in the introduction. How developed NPs can or cannot overcome this resistance mechanism?

Reply: We thank the reviewer for raising this question. We have elaborated on it to be clearer about the implications of developed NPs in overcoming the resistance mechanism. Information regarding this aspect was included between lines in the introduction section in the following paragraph (lines 109-129): “PDT of melanoma is limited also by resistance mechanisms driven mainly by elevated levels of melanin. These include optical interference by the highly-pigmented mela-nin, the antioxidant effect of melanin, the sequestration of PS inside melanosomes, defects in apoptotic pathways, and the efflux of PS by ATP-binding cassette (ABC) transporters [13].

Nanoparticles (NPs) have shown a potential to overcome some of these challenges and increase the PDT efficacy [14] . For instance, NPs may deliver and concentrate PS in the cytoplasm of cancer cells thus allowing saturation of ABC transporters [15].

Due to their properties, TiO2 NPs may represent a promising candidate for PDT [16]. However, their use in this application is constrained by low tissue penetration of ultravio-let light and the harmful effects of ultraviolet radiation. To increase ROS formation and to improve the physicochemical properties, particularly the absorption of visible light, these NPs are conjugated with PS such as porphyrins [17,18]. By electron transfer from the pho-tosensitizer excited by visible light to the TiO2 NPs, ROS can be produced. In this way, an increase in photodynamic activity is achieved by the synergistic production of ROS by both, PS and TiO2 [19]. This may help elevate the oxidative stress in melanoma cells con-sidering that melanin may act as a ROS scavenger contributing to resistance [20].  More, TiO2 NPs have been proven to be effective in preventing drug efflux caused by multidrug resistance (MDR), which enables their selective accumulation at the tumor site [21]. Others have indicated that TiO2 NPs increase the selectivity of porphyrin-based PS and reduce their adverse effects [16]. “

The statement “Photodynamic therapy (PDT) is an anticancer strategy developed in the last century for the treatment of melanoma…” is not true.

Reply: We thank the reviewer for this observation and we agree that this wording led to a false affirmation. The phrase was reformulated as follows (lines 64-67): “Photodynamic therapy (PDT) is an anticancer strategy developed in the last century for the treatment of several types of cancer. [4]. The research for use of PDT as a potential treatment option for melanoma was started in 1988 with two significant studies [5,6] that laid the foundation for the subsequent investigation of PDT for this type of cancer [7].”

  1. d)     The discussion is also too long and basically repeats the results and the introduction. Please, reorganize it and answer the question- “What is the mechanism behind the photodynamic activity enhancement when TiO2 NPs are conjugated with TMPyP4?”

Who absorbs the light (NPs or PSs) and what happens next? I assume, there is SET involved in the process, but I don’t clearly understand who generates ROS in the end. 

Reply: Accordingly, to the reviewer’s comment, the discussion section was reorganized and we introduced the following text in the manuscript (lines 119-124): “To increase ROS formation and to improve the physicochemical properties, particularly the absorption of visible light, these NPs are conjugated with PS such as porphyrins [17,18]. By electron transfer from the photosensitizer excited by visible light to the TiO2 NPs, ROS can be produced. In this way, an increase in photodynamic activity is achieved by the synergistic production of ROS by both, PS and TiO2 [19].”

We also deleted from Discussion section rows 610-613, 615-617 and 657-666, which have information available in the Results section.

  1. e)     In the discussion, please compare other NPs proposed by your group or other groups with novel NPs and give several statements on how you see further development of the field.

Reply: We thank the reviewer for his kind suggestion, we added in the Discussion Section the paragraph (lines 581-596): “TiO2 NPs are prevailingly used in tissue and bone engineering due to their capacity to promote cell migration and adhesion [51,52], and these NPs serve as antibacterial agents through their ability to produce ROS in the presence of UV light [53,54]. Furthermore, they also can exhibit anticancer effects via intracellular ROS production [55]. Compared with iron oxide NPs (Fe2O3 NPs), which are mostly used as compounds-delivery systems for anticancer treatments by loading the NPs to target a specific binding protein or to control the nano-complexes with an external magnetic field [56,57], TiO2 NPs are not only a de-livery vector, it has also a photocatalytic activity (generating ROS species) which moreover can be improved by functionalization with an appropriate sensitizer, such as porphyrins and phthalocyanines [58–61].

Currently, various types of nanoparticles (zinc, cerium, iron, silver and titanium ox-ide NPs) are promising candidates for biomedicine and in the future, nanotechnology may become the most used method in clinical practice, with a considerable amount of research data in vitro and in vivo studies, due to the small size properties and large surface area, that give scientists new tools for identifying and treating diseases that were previously thought to be unapproachable as a consequence of size restrictions. “

2)      “Taking into consideration the absorbance of 0.48 for the TMPyP4/TiO2 suspension at 422 nm (the background due to NPs diffusion subtracted), we can infer a concentration of 3 µg/mL for TMPyP4 in the TMPyP4/TiO2 complexes, corresponding to 90µg/ml of TiO2.”

Could you elaborate on how you measure these concentrations?

Reply: We extend the text as follows (lines 390-398): “The intensity of the 422 nm absorption peak of TMPyP4 was used to quantify the loading of porphyrin in the TMPyP4/TiO2 complexes. From Figure 4, it can be extracted for the complexes, a 0.48 value for the absorbance corresponding to this peak (the background due to NPs diffusion being subtracted). Taking into account that for a known concentration of 10 µg/mL, the absorbance of TMPyP4 at 422 nm is 1.6, it can be estimated in complex samples a loading of 3 µg/mL for the porphyrin. Similarly, by considering the absorbance of TiO2 suspension for 120 µg/mL (Figure 4) and that of TMPyP4/TiO2 at 700 nm (where no features of TMPyP4 are present), we can deduce a 90 µg/mL concentration of TiO2 in the complexes.”

In the manuscript you mentioned that aggregates are formed in solutions. How this fact affects such measurements?

Reply: We explained this by adding the following paragraph in Discussions, lines 650-655. “Considering that at concentration of 10 µg/mL, no dimerization of TMPyP is observed (specific absorption bands of dimers are not present), we think that aggregation does not affect these measurements. Regarding NPs, similar aggregates are observed in the SEM images for TiO2 and TMPyP/TiO2, so, also in this case the absorption spectra should give a measure for the amount of TiO2 in the samples.”

3)     It would be better to evaluate ROS generation yields rather than phosphorescence, since it would give quantitative results. If its possible, I would suggest doing that.

Reply: We agree with the reviewer’s comment. Unfortunately, we do not have compounds for spectrophotometric detection of ROS. By measuring the phosphorescence of singlet oxygen generated, we could give an indication of this species formation. We intend to acquire specific compounds for ROS quantitative determination.

4)     I think, TMPyP4 can easily absorb fluorescence of H2DCFDA, is it an issue? How accurate are such studies (Figure 9)?

Reply: Theoretically, the TMPyP4 being a cationic porphyrin could interact with negatively charged DCF molecules through electrostatic interactions and form a complex. This complex is no longer able to emit fluorescence, which could result in a decrease in fluorescence signal. However, in practice, many factors may intervene including the concentration of compounds, conjugation reactions, detection sensitivity, etc. Looking at the result from figure 9, it is obvious that the fluorescence was not quenched in the case of TMPyP4 because it showed increased production of ROS level at a dose of 1 μg/mL. Moreover, in several previous studies of ROS generation by TMPyP4 no interference with the DCFDA assay was reported. [Zhuang, Xin-Ying; Yao, Yong-Gang (2013). Mitochondrial dysfunction and nuclear-mitochondrial shuttling of TERT are involved in cell proliferation arrest induced by G-quadruplex ligands. FEBS Letters, 587(11), 1656–1662.; Shi T, Wang M, Li H, Wang M, Luo X, Huang Y, Wang HH, Nie Z, Yao S. Simultaneous Monitoring of Cell-surface Receptor and Tumor-targeted Photodynamic Therapy via TdT-initiated Poly-G-Quadruplexes. Sci Rep. 2018 Apr 3;8(1):5551.; Chen, J., Jin, X., Shen, Z. et al. H2O2 enhances the anticancer activity of TMPyP4 by ROS-mediated mitochondrial dysfunction and DNA damage. Med Oncol 38, 59 (2021)].

5)     The way I see it, toxicity and water solubility predictions are not well-justified. You have your NPs in hand. Why do you need to study them through computations?  TMPyP4 has been extensively studied in vivo (Mol Cancer Ther (2002) 1 (8): 565–573) and its toxicity profile is known. Also, I believe that solubility and toxicity of TMPyP4 will be different after immobilization on NPs. 

Reply: We thank the reviewer for bringing this to our attention. Mathematical models determine the toxicity measures of a compound based on the physical properties of chemical molecules. Computational toxicity prediction is useful in the early stages of drug development to identify compounds that are likely to fail in subsequent studies. We were unable to identify all of the toxicity characteristics predicted in this study in other "in vitro" investigations; for example, AMES toxicity and hERG I and II inhibitory activity reports are unavailable; therefore, in this study, we added the toxicity predictions that will fulfill TMPyP's toxicological profile. However, we have removed the water solubility predictions from the Table 3 (line 469).

Round 2

Reviewer 1 Report

The authors have addressed most of my concerns, and I have no more questions.

Author Response

Review #1

The authors have addressed most of my concerns, and I have no more questions.

Reply: We sincerely thank reviewer 1 for the attention granted to our manuscript and for all remarks which contributed to the improvement of our manuscript.

Reviewer 2 Report

Dear Authors,

Thank you for your comments and revising the manuscript. However, I still have several points I would like to discuss with you, these are following:

1.      Unfortunately, I am not convinced with your responses regarding benefits of NPs/photosensitizers formulation for the melanoma treatment.

If the pigment harvests photons under blue light, it means that no matter how efficient your photosensitizer is, there will be a poor outcome. This can be observed using only in vivo models. Please, consider the absorption spectra of melanin https://www.cl.cam.ac.uk/~jgd1000/melanin.html and give references for previous studies conducting PDT of melanomas under blue light in vivo. Are those studies successful? What do you expect from your NPs?
In provided review 10.21037/qims.2018.05.04 [ref 14], only one NP/PS system was proved to work for melanomas and it was activated in the area where melanin is transparent (https://www.nature.com/articles/nm.2933).

“For instance, NPs may deliver and concentrate PS in the cytoplasm of cancer cells thus allowing saturation of ABC transporters [15].”

“More, TiO2 NPs have been proven to be effective in preventing drug efflux caused by multidrug resistance (MDR), which enables their selective accumulation at the tumor site [21].”

In both studies targeting vectors were incorporated into NPs, but your molecules lack them, so, I don’t think that its relevant.

2.      “…that give scientists new tools for identifying and treating diseases that were previously thought to be unapproachable as a consequence of size restrictions

What are “size restrictions” in this context?

3.      Please, give references for the method you used when assessing of NP’s concentrations. Have you done calibration curves? Do they obey the Beer-Lamber law?

Considering that at concentration of 10 µg/mL, no dimerization of TMPyP is observed…” requires a citation.

4.      We focused on TMPyP4 toxicity since the molecule's absorption and distribution may change because of the NPs delivery” . That said, you calculate toxicities for TMPyP alone, doesn’t make sense to me.

5.      I have to agree with the reviewer 1 on the permeability issue. Please, evaluate localization of NPs in cells by means of, for example, fluorescence spectroscopy. It is no clear if synthesized nanoobjects penetrate cell’s membrane. 

Author Response

Review #2

Dear Authors,

Thank you for your comments and revising the manuscript. However, I still have several points I would like to discuss with you, these are following:

Reply: We thank the reviewer for the attention granted to our paper. We sincerely appreciated all the remarks and we feel that the resubmitted manuscript which was revised according to the reviewer’s requests was considerably improved.

  1. Unfortunately, I am not convinced with your responses regarding benefits of NPs/photosensitizers formulation for the melanoma treatment.

    If the pigment harvests photons under blue light, it means that no matter how efficient your photosensitizer is, there will be a poor outcome. This can be observed using only invivo models. Please, consider the absorption spectra of melanin https://www.cl.cam.ac.uk/~jgd1000/melanin.html and give references for previous studies conducting PDT of melanomas under blue light in vivo. Are those studies successful? What do you expect from your NPs?
    In provided review 10.21037/qims.2018.05.04 [ref 14], only one NP/PS system was proved to work for melanomas and it was activated in the area where melanin is transparent (https://www.nature.com/articles/nm.2933).

Reply: We thank the reviewer for this comment. The only solutions suggested in the literature to the problems linked to the absorption of light by melanin in PDT were depigmentation or suppression of melanogenesis (tyrosinase activity). [Li XY, Tan LC, Dong LW, Zhang WQ, Shen XX, Lu X, Zheng H, Lu YG. Susceptibility and Resistance Mechanisms During Photodynamic Therapy of Melanoma. Front Oncol. 2020 May 12;10:597]. This strategy was considered in several in vitro studies which support the hypothesis that combining the inhibition of melanogenesis with PDT should be explored as a valid therapeutic target for the management of melanoma.

  1. Sharma KV, Bowers N, Davids LM. Photodynamic therapy-induced killing is enhanced in depigmented metastatic melanoma cells. Cell Biol Int. 2011 Sep;35(9):939-44.
  2. Sharma KV, Davids LM. Depigmentation in melanomas increases the efficacy of hypericin-mediated photodynamic-induced cell death. Photodiagnosis Photodyn Ther. 2012 Jun;9(2):156-63. doi: 10.1016/j.pdpdt.2011.09.003.

However, this strategy was applied also in vivo:

  1. Ma LW, Nielsen KP, Iani V, Moan J. A new method for photodynamic therapy of melanotic melanoma -- effects of depigmentation with violet light photodynamic therapy. J Environ Pathol Toxicol Oncol. 2007;26(3):165-72

Our NP complexes in their current form do not have the capacity to overcome the melanin issue maybe if combined with an agent with the ability to inhibit melanin production or induce depigmentation or used in vivo after photobleaching of melanin. More, the cell model used in our study was an amelanotic melanoma cell line – Mel-Juso which comes in response to the previously mentioned. The amelanotic melanoma cells used in our study represent a model of depigmented melanoma in which melanin interference is not involved. The advantage of synthesized complex in comparison to free PS is linked only to the increased cytotoxicity in cancer cells vs normal cells and enhancement of overall production of ROS by a synergic activity.

Some sentences were reviewed in the manuscript (lines 84-90, 119, 192, 512, 518, 519, 690) for clarification.

“For instance, NPs may deliver and concentrate PS in the cytoplasm of cancer cells thus allowing saturation of ABC transporters [15].”

“More, TiO2 NPs have been proven to be effective in preventing drug efflux caused by multidrug resistance (MDR), which enables their selective accumulation at the tumor site [21].”

In both studies targeting vectors were incorporated into NPs, but your molecules lack them, so, I don’t think that its relevant.

Reply: We thank the reviewer for this observation. We agree that the provided references were not good examples. However, the potential capacity of TiO2 NPs to inhibit the P-gp-mediated MDR was shown also in other studies utilizing TiO2 NPs lacking targeting vectors. For example, Song et al., 2006 explored the effect of nano TiO2 conjugated with the anticancer drug daunorubicin to inhibit the drug resistance of leukemia K562 cells. Their results indicate that the relative formation of ion-pair as well as the competitive binding of TiO2 NPs and daunorubicin molecules to the overexpression P-gp on the cell membrane of K562 cells increase the accumulation of daunorubicin into cells. The authors suggested that TiO2 NPs could play an important role to facilitate the drug uptake of cancer cells and may also act as an effective anti-MDR agent to inhibit relative drug resistance. [Min Song; Renyun Zhang; Yongyuan Dai; Feng Gao; Huimei Chi; Gang Lv; Baoan Chen; Xuemei Wang (2006). The in vitro inhibition of multidrug resistance by combined nanoparticulate titanium dioxide and UV irradition. , 27(23), 4230–4238 – reference 22]. 

These sentences were reviewed for clarity as follows (lines 84-90, 100-102):

“Therefore, depigmentation or suppression of melanogenesis (tyrosinase activity) have been used as strategies to address the problems linked to melanin interference in PDT [14].

Nanoparticles (NPs) have shown the potential to improve PDT efficacy. For instance, NPs may deliver and concentrate PS in the cytoplasm of cancer cells and increase the PS toxicity by providing a source of supplementary ROS [15].”

“More, under UV irradiation, TiO2 NPs have been proven to be effective in preventing drug efflux caused by multidrug resistance (MDR), which enables enhanced drug accumulation in cells [22].”

  1. “…that give scientists new tools for identifying and treating diseases that were previously thought to be unapproachable as a consequence of size restrictions”

What are “size restrictions” in this context?

Reply: We thank the reviewer for this observation. We agree that this wording was unclear, in fact, we wanted to refer to the conventional drug administration system. Not to be confusing we deleted “as a consequence of size restrictions” (line 547).

  1. Please, give references for the method you used when assessing of NP’s concentrations. Have you done calibration curves? Do they obey the Beer-Lamber law?

Reply: We thank the reviewer for this observation. We have done calibration curves for TMPyP and also for TiO2 NPs. We modified Figure 4 in the revised manuscript (we introduced an inset with the calibration curve for TmPyP4) and also we revised the text as follows (lines 341-354): 

“The inset of Figure 4 shows the calibration curve consisting in the absorbance values for the peak at 422 nm versus concentration for several TMPyP4 solutions in water with concentrations in the range 1-10 µg/mL. Taking into account the 0.47 absorbance for TMPyP4/TiO2 suspension (Figure 4) at 422 nm nm (the background due to NPs diffusion being subtracted), we can extrapolate from the calibration curve a TMPyP loading of 3 µg//mL in the complexes. From Figure 4, it can be extracted for the complexes, a 0.47 value for the absorbance corresponding to this peak (the background due to NPs diffusion being subtracted). Taking into account that for a known concentration of 10 µg/mL, the absorbance of TMPyP4 at 422 nm is 1.6, it can be estimated in complex samples a loading of 3 µg/mL for the porphyrin.  Similar calibration curves were used for TiO2 NPs (data not shown), by considering the absorbance of TiO2 suspension for 0.12 mg/mL (Figure 4) and that of TMPyP4/TiO2 at 700 nm (where no features of TMPyP4 are present), we can deduce a 0.1 mg/mL concentration of TiO2 in the complexes.”

<<figure>>

Figure 4. The absorption spectra for 0.01mg/ml TMPyP4 water solution, 0.12mg/ml TiO2 water suspension, and TMPyP4/TiO2 complexes water suspension. The inset shows the calibration curve for determination of TMPyP4 concentration in the complexes.

Regarding TiO2 NPs concentration, we did not shown the data in the manuscript, we exhibited here the graphs to show that we have also a good linearity with concentration:

<<figure>>

“Considering that at concentration of 10 µg/mL, no dimerization of TMPyP is observed…” requires a citation.

Reply: In fact, now, it can be seen from the behavior of absorption with concentration up to 10 µg/mL, we changed the text as follows (lines 592-594):

Considering that at concentrations up to 10 µg/mL, no dimerization of TMPyP is observed (the calibration curves for TMPyP determination based on monomer-specific absorption peak at 422 nm showed a good linearity…

We also insert only here the graph with spectra for TMPyP for several concentrations, where no additional dimers bands are formed on the right side of the monomer peak:

<<figure>>

  1. “We focused on TMPyP4 toxicity since the molecule's absorption and distribution may change because of the NPs delivery” . That said, you calculate toxicities for TMPyP alone, doesn’t make sense to me.

Reply: We thank the reviewer for this observation, we have deleted the sentence from the text (lines 406-408):

"We focused on TMPyP4 toxicity since the molecule's absorption and distribution may change because of the NP's delivery ".

We believe that the TMPyP toxicity predictions are relevant to the study because they may help to close a knowledge gap in this area. Unfortunately, the toxicity of the complex cannot be predicted due to the nanoparticle size. Therefore, we only make predictions about the toxicity of porphyrin.

  1. I have to agree with the reviewer 1 on the permeability issue. Please, evaluate localization of NPs in cells by means of, for example, fluorescence spectroscopy. It is no clear if synthesized nanoobjects penetrate cell’s membrane. 

Reply: We agree with the reviewer’s comment, and perfectly understand the necessity of this evaluation. Unfortunately, at this moment we don’t have access and expertise to perform electronic or confocal microscopy to reveal the internalization of our NPs complexes. With all respect, we don’t think fluorescence spectroscopy will clarify the localization of NPs because as far as we know it cannot distinguish between the cell-uptaken and membrane-bound NPs.

Round 3

Reviewer 2 Report

Dear Authors,

Thank you for your responses! I think we resolved almost all problems I had been concerned about. 

As for the permeability issue, I encourage you to find collegues familiar with confocal microscopy of fluorescent dyes. TMPyP is a fluorescent compound which makes your NPs suitabale objects for localization studies. If you want to level up quality of the research work, that point should be adressed in the future.